# Association of Total and Trimester-Specific Gestational Weight Gain Rate with Early Infancy Weight Status: A Prospective Birth Cohort Study in China

**DOI:** 10.3390/nu11020280

**Published:** 2019-01-28

**Authors:** Jiajin Hu, Izzuddin M. Aris, Emily Oken, Yanan Ma, Ning Ding, Ming Gao, Xiaotong Wei, Deliang Wen

**Affiliations:** 1Department of Social Medicine, School of Public Health, China Medical University, Shenyang 110122, Liaoning, China; jjhu@cmu.edu.cn (J.H.); cmugaom@163.com (M.G.); xtwei@cmu.edu.cn (X.W.); 2Division of Chronic Disease Research Across the Lifecourse, Department of Population Medicine, Harvard Medical School, Boston, MA 02215, USA; Izzuddin_Aris@harvardpilgrim.org (I.M.A.); emily_oken@harvardpilgrim.org (E.O.); 3Department of Obstetrics and Gynecology, Yong Loo Lin School of Medicine, National University of Singapore, 119228 Singapore, Singapore; 4Agency for Science, Technology and Research, Singapore Institute for Clinical Sciences, 119228 Singapore, Singapore; 5Department of Nutrition, Harvard T.H. Chan School of Public Health, Boston, MA 02113, USA; 6Department of Epidemiology and Health Statistics, School of Public Health, China Medical University, Shenyang 110122, Liaoning, China; ynma@cmu.edu.cn; 7Department of Curriculum and Teaching Research, Research Center of Medical Education, China Medical University, Shenyang 110122, Liaoning, China; nding@cmu.edu.cn

**Keywords:** gestation weight gain, trimester-specific, infancy, weight status, body mass index

## Abstract

Studies to examine the associations of gestational weight gain (GWG) with offspring weight status during infancy are needed, especially among Asian populations. We examined 801 mother–infant pairs from a prospective cohort study in China to assess the associations of total and trimester-specific GWG with early infant weight status. We used linear mixed effects models and linear regression models to examine the longitudinal and time-point associations of GWG rate (kg/week) with infant growth measures (z-scores of body-mass-index (BMIZ), weight-for-age (WFAZ) and length-for-age (LFAZ)) at birth, 1, 3, and 6 months. Greater total GWG rate was associated with higher BMIZ (β:1.34 SD units (95% CI: 0.84, 1.83) per 1 kg/week increase in GWG) and higher WFAZ (β:1.18 SD units (95% CI: 1.01, 2.28)) across the first 6 months of life. GWG rate in the first two trimesters but not in the third trimester was positively associated with infant BMIZ. The association between GWG rate and infant BMIZ was significant at all timepoints and more pronounced in normal weight women and among male infants. In conclusion, greater GWG rate is positively associated with offspring BMIZ in the first 6 months of life, the association is mostly driven by GWG in the first two trimesters.

## 1. Introduction

The epidemic of childhood obesity is a global challenge [1]. In China, the prevalence of childhood obesity has doubled over the past few decades [2]. While existing evidence has shown that greater gestational weight gain (GWG) is associated with larger offspring birth size [3] and higher body mass index (BMI) in childhood, adolescence, and adulthood [4,5,6,7,8,9,10,11,12], few studies [11,13,14,15] have examined gestational weight gain with offspring weight status in the first year of life, a sensitive period that contributes to increased risk of obesity later in life [16,17,18,19].

Furthermore, the effects of GWG programing childhood weight status may vary according to the timing of weight gain during pregnancy. First and second trimester weight gain primarily represents maternal fat deposition, which may increase the placental transfer of nutrients from mother to fetus, while third trimester weight gain reflects fetal growth which may influence fetal body composition [20]. Most previous studies found that early, rather than late GWG, was associated with offspring weight status [14,21,22,23,24,25], however other studies have suggested that excessive late weight gain may also contribute to offspring risk for obesity [26,27]. Most of these studies were based in Western populations. Although a previous study in China [28] has examined associations of trimester specific GWG with offspring weight status at birth, further studies are needed to assess this association in Asian populations, where GWG patterns may differ from Western populations [20,29,30].

In addition, existing studies have reported sex-specific effects of pre- and postnatal exposures such as pre-pregnancy weight status [31], smoking during pregnancy [32,33], and breastfeeding [34] on offspring obesity. To our knowledge, few studies have identified sex-specific effects of GWG on child weight status after birth. Characterization of these associations will help to better define the target population and opportunities for inventions to prevent lifetime risk of obesity.

To address these gaps, we investigated associations of total and trimester-specific GWG rate with offspring weight status during early infancy in a prospective pre-birth cohort study in China, the Born in Shenyang Cohort Study (BISCS). We hypothesized that: (1) greater GWG rate would be positively associated with weight status of infant; (2) early pregnancy (first and second trimesters) GWG would have stronger associations with infant weight status; (3) the association of GWG with infant weight status would differ by infant sex or pre-pregnancy women weight status.

## 2. Materials and Methods

### 2.1. Study Population and Design

We established a multicenter prospective pre-birth cohort, “Born in Shenyang Cohort Study (BISCS)”, to examine associations of prenatal factors in relation to maternal and child health. Between April and September 2017, we enrolled healthy single pregnant women at 21–24 weeks gestation at 54 hospitals and community healthcare centers which are all the maternity clinics for antenatal visit in the urban area of Shenyang, northeast of China. We included women with singleton pregnancies, gestational age <24 weeks, and who had no plans to move away in the next three years. Among 2068 women, 1338 agreed to participate in the study and 1260 had a live singleton birth (Figure 1). All mothers provided written informed consent, and the ethics committee of the China Medical University approved the study.

We conducted in-person visits with mothers during the second trimester (mean ± SD: 22 ± 1.2 week) and third trimester (mean ± SD: 34.5 ± 1.9 week). We also conducted home visits with mothers and infants within 7 days after birth, and research visits at the child development clinics at ages 1, 3, and 6 months. We collected socio-demographic, environmental, behavioral, and clinical information from mothers and children via questionnaires and medical records, and performed anthropometric measurements.

Among 1260 women with singleton live births, 934 mother–infant pairs responded at 1 month visit (Appendix A). The respond rates of 1, 3, and 6 months visit were 74.1%, 72.9%, and 72.2%, respectively. We further excluded women with pre-gestational diabetes mellitus (*n* = 3) or delivery at <34 weeks of gestation [22] (*n* = 5). Of the 926 eligible mother-child pairs, we further excluded from the present analysis, those without data on gestational weight, or any weight records within 4 weeks preceding delivery (*n* = 125). A total of 801 mother-child pairs were included in the present study (Appendix A). We compared characteristics of the 801 participants included in the present analysis with participants excluded. We found similar characteristics between two groups except that women included in the analysis reported lower smoking rates before pregnancy (4.2% vs. 6.0%).

### 2.2. Exposure: Gestational Weight Gain

We defined total GWG as the difference between self-reported pre-pregnancy weight and last recorded weight before delivery. We obtained a median of 10 (range 5 to 17) weight records over the entire duration of pregnancy. We calculated GWG rate (kg/wk) as total GWG divided by the number of gestational weeks at delivery. We defined the first two trimesters as the duration from the date of the last menstrual period to day 196 (28 weeks) [35] of pregnancy and the third trimester as day 196 of pregnancy to delivery. We performed linear interpolation between the two closest weight measures to estimate weights at day 196 [36] if the weight was not measured at day 196. The GWG rate in the first two trimesters was calculated as weight gain during the first two trimesters divided by 28, and the third trimester GWG rate was calculated as weight gain during the third trimester divided by number of weeks from the 28th week of gestation to delivery.

### 2.3. Outcome: Child Anthropometry

We extracted weight and length at birth from medical records. Trained pediatricians measured the child’s length and weight during research visits at ages 1, 3, and 6 months using calibrated infant stadiometers (Seca 416; Seca Corporation, Hamburg, Germany) and weighing scales (Seca 376+; Seca Corporation, Hamburg, Germany). We measured length and weight twice to the nearest 0.1 cm and 0.01 kg respectively, and calculated the mean of the two measurements.

We calculated offspring BMI by dividing infant weight in kilograms by the square of height in meters measured at birth and 1, 3, 6 months of age. We calculated age- and sex-specific z-scores for BMI (BMIZ), weight (WFAZ), and length (LFAZ) based on the World Health Organization (WHO) child growth reference [37].

### 2.4. Assessment of Covariates

We measured mother’s height at the initial prenatal visit using a calibrated stadiometer (Seca 217; Seca Corporation, Hamburg, Germany). Mother’s pre-pregnancy BMI was calculated by dividing self-reported pre-pregnancy weight in kilograms by the square of measured height in meters. Father’s BMI was calculated on self-reported height and weight. Both mother’s pre-pregnancy BMI and father’s BMI were categorized as underweight (BMI <18.5 kg/m^2^), normal-weight (18.5 kg/m^2^ ≤ BMI < 23 kg/m^2^) or overweight/obese (BMI ≥23 kg/m^2^) using WHO references for Asian population [38]. We combined overweight and obese women into a single category because of a limited sample size. We collected information on maternal age, education, race, household income, and smoking status during and before pregnancy using interviewer-administered questionnaires. We also collected prenatal clinical information from the medical records of mothers, including parity and glucose tolerance. During infancy, we collected information on the mode of infant feeding, infant’s exposure to environmental tobacco smoke, infant’s television viewing status, and sleeping hours per day at ages 1, 3, and 6 months respectively using questionnaires. We categorized the mode of infant feeding at ages of 1, 3, and 6 months as: exclusive breastfeeding, mixed breast and formula feeding, exclusive formula feeding, or weaned from breastfeeding [13].

### 2.5. Statistical Analysis

We used ANOVA and chi-square tests to describe the characteristics of exposures and covariates according to quartiles of total GWG rate.

We examined the longitudinal associations of total and trimester-specific GWG rate (kg/week) with infant sex- and age- specific z-score for BMI, weight and length from birth to 6 months using linear mixed effects (LME) models, which takes into account within-subject correlation of repeated measurements and at the same time allows for incomplete outcome measurement [39]. We fitted the models using an unstructured covariance matrix for random effects variables (intercept and slope) and used maximum likelihood estimation method. We conducted crude and adjusted analyses: Model 1: adjusted for linear, quadratic and cubic terms for infant’s age to estimate the associations of GWG rate (per kg/week) with growth measure z-scores in the first 6 months; Model 2: Model 1 + pre-pregnancy BMI; Model 3: Model 2 + maternal age, race, parity, educational attainment, household income, smoking status, and paternal BMI. In the third trimester analysis, we additionally adjusted for GWG rate in the first two trimesters and gestational diabetes mellitus status.

We also used multivariable linear regression models to examine associations of total and trimester-specific GWG rate as well as the pre-pregnancy weight status with infant sex- and age-specific z-scores for BMI, weight and length at each time point (at birth and at ages 1, 3, and 6 months). The full model adjusted for pre-pregnancy BMI, maternal age, race, parity, educational attainment, household income, smoking status, and paternal BMI. We tested interactions of GWG rate with pre-pregnancy BMI category and infant sex in relation with infant growth measures by including corresponding interactions into the models. Furthermore, we performed stratified analyses to investigate the association of GWG rate with infant size among women of different pre-pregnancy weight status. In addition, we also stratified analyses according to infant sex to identify sex-specific associations of total and trimester-specific GWG rate with infant growth.

For sensitivity analyses, we used weight-for-length z-scores (based on the WHO child growth reference) as outcomes to be compared with BMI z-scores as outcomes in the LME analysis. We also adjusted for infant variables that might mediate associations of GWG with outcomes, including mode of infant feeding, television viewing status, and secondhand smoke exposure at the age of 1 month.

To get a full sample, multiple imputation was used to fulfill the missing values in linear regression models. We performed all analyses using Stata S.E. version 13 (Stata Corp, TX, USA). A two-side *p*-value <0.05 was considered statistically significant.

## 3. Results

### 3.1. Maternal Characteristics

There were no significant differences in demographic (maternal age, race, educational attainment, household income per year, and smoking status) or clinical characteristics (parity, gestational length, paternal BMI, offspring sex) across quartiles of total GWG (Table 1). Compared to women in the lowest GWG rate quartile, those in the highest quartile had greater GWG rate during both the first two trimesters (0.50 vs. 0.19 kg/wk) and the third trimester (0.93 vs. 0.50 kg/wk), greater offspring birth weight (3548 vs. 3292 g), lower prevalence of gestational diabetes mellitus (15.5% vs. 28.9%), and lower prevalence of pre-pregnancy overweight/obesity (30.0% vs. 47.9%).

### 3.2. Longitudinal Analysis (LME Models)

Table 2 shows the associations between total and trimester-specific GWG rates and infant growth measure z-scores from birth to 6 months assessed using LME models. After adjusting for covariates (Model 3), greater total GWG rate was associated with higher BMI z-score (β:1.34 SD units (95% CI: 0.84, 1.83)) and weight-for-age z-score (β:1.18 SD units (95% CI: 1.01, 2.28)), but not associated with length-for-age z-score. Similar results were reported when assessing associations of first two trimesters GWG rate with infant growth measures. Greater first two trimesters GWG rate was associated with higher BMI z-score (β:1.31 SD units (95% CI: 0.85, 1.77)) and weight-for-age z-score (β:1.08 SD units (95% CI: 0.76, 1.49)), but not associated with length-for-age z-score. Third trimester GWG rate was not associated with BMI z-score, weight-for-age z-score, or length-for-age z-score in all models. In sensitivity analysis, using infant WFLZ as outcomes comparing with BMI z-score, or further adjusted infant variables did not change the results (Appendix A). Pre-pregnancy BMI was also positively associated with infancy BMIZ independent of GWG rate (Appendix A).

### 3.3. Individual Time-Points Analysis (Multipule Linear Regression Analysis)

Table 3 describes time-point associations of total and trimester-specific GWG rate with infant growth measure z-scores at birth, at age of 1, 3, and 6 months. Greater total GWG rate was associated with higher BMI z-score at all time points. The effect estimate was highest at birth (1.56 SD units (0.91, 2.11)), and decreased at age of 1 month (0.86 SD units (0.91, 2.11)), then remained relatively stable at age of 3 months (0.99 SD units (0.40, 1.59)), and age of 6 months (0.82 SD units (0.16, 1.47)). Similar results were observed when using weight-for-age z-score as outcome. Total GWG rate was positively associated with length-for-age z-score only at birth (0.78 SD units (0.09, 1.47)), but not at later time during early infancy. First two trimester GWG rate was positively associated with infants’ BMI z-score and weight-for-age z-score at all four time points, but not associated with length-for-age z-score at any time point. GWG rate in the third trimester was not associated with infant BMI z-score, weight-for-age z-score, or length-for-age z-score at any age.

### 3.4. Associations of Total GWG Rate with Early Infancy Sex- and Age-Specific z-Scores for BMI by Pre-Pregnancy Weight Status

In stratified analyses by pre-pregnancy weight categories (Figure 1), we found that the associations of total GWG rate with early infancy BMI z-score were different within different pre-pregnancy weight categories (*p* for interaction <0.05). In normal weight women, total GWG rate was positively associated with offspring BMI z-score at birth, 1 month, 3 months, and 6 months. In underweight women, total GWG rate was positively associated with offspring BMI z-score only at age of 3 months. No associations were observed between total GWG rate and offspring BMIZ in overweight/obese women.

### 3.5. Associations of Total and Trimester-Specific GWG Rate with Early Infancy BMI z-Score by Infant Sex

Among male infants, total GWG rate and GWG rates in the first two trimesters were associated with infant BMI z-score at birth, 3, and 6 months (Figure 2). However, among female infants, total GWG rate and GWG rates in the first two trimesters was associated with BMI z-score only at birth and 1 month, but not at later ages. GWG rate in the third trimester was not associated with infant BMI z-score in male or female infants at any timepoint. In both sexes, total GWG rate and GWG rates in the first two trimesters had nearly identical associations with BMI z-score across early infancy (Figure 2).

## 4. Discussion

In this prospective birth cohort study in China, we found that the rate of gestational weight gain was positively associated with offspring BMI z-score across early infancy. The association was largely driven by GWG in the first two trimesters rather than the third trimester. The observed associations were more pronounced in normal weight women and in male rather than female infants.

Our findings provide additional evidence to the existing literature on the relationship between trimester-specific GWG and offspring weight status in the early stages of life. Although a previous study examined the association between trimester specific GWG with infant birthweight in the Asian population, our study is the first study to assess associations of trimester-specific GWG on offspring weight status across early infancy in Asia. Our findings may help develop intervention strategies focused on controlling the rapidly increasing prevalence of childhood obesity in Asia.

Our results are consistent with previous studies [11,12,24] that showed associations of greater total GWG rate with higher offspring BMI. A possible mechanism could be that greater GWG reflects overnutrition in utero, thereby exposing the fetus to more glucose and fatty acids [20,40], which could increase fetal secretion of insulin and further lead to adiposity. However, in stratified analyses, only among normal weight women, the associations of GWG rate with offspring BMI z-score were statistically significant across early infancy, which is in contrast with a previous study reporting that the association between offspring obesity and GWG is stronger among mothers who were overweight prior to pregnancy than those who had normal weight [41]. One possible mechanism could be that participants who were overweight before pregnancy also gained less weight. Therefore, the smaller effects of GWG in overweight participants may be due to the fact that they gained less weight in general. Furthermore, overweight women may generally have heavier children, leading to a ceiling effect and therefore no relation between GWG and child BMI in overweight mothers. Interventions to prevent excessive GWG, and thus subsequent childhood obesity, in normal weight pregnant women may actually be more effective than interventions in underweight or overweight/obese women. According to our findings, weight-for-age z-score but not length-for-age z-score was associated with GWG rate, which indicates the positive association of GWG on BMI z-score was mainly driven by the weight but not the length of children.

Our findings confirmed previous studies [21,22,25] that reported first and second trimester GWG, but not third trimester gain, were associated with offspring BMI. For example, in the Danish National Birth Cohort, Anderson et al., found that average weekly GWG rates in first and second trimesters were positively associated with child’s BMI at 7 years [25]. In the US Project Viva cohort, Hivert et al., found that greater GWG rate in the first two trimesters, was associated with child’s BMI z-score and fat mass at 7.9 years [22]. Both these studies found no association of third trimester GWG with mid-childhood adiposity outcomes. Our research also aligns with findings of Karachaliou et al. [14] in that first trimester GWG rate, but not the second and third trimesters combined, was associated with higher BMI z-score in 6-month-old infants. However, that study [14] combined second and third trimesters, which may mask the effect of second trimester GWG rate. Other studies however, have suggested that late pregnancy weight gain also may influence childhood adiposity risk [26,27], although these studies did not isolate the third trimester as an exposure period.

Taken together, these findings suggest that the early pregnancy period is a highly sensitive period for fetal adiposity programing, consistent with the fact that fetal adipose tissue begins to develop around 14–16 weeks of gestation [42]. Greater GWG in early pregnancy may reduce maternal insulin sensitivity and glucose tolerance which could expose the fetus to an increased glucose supply [43,44] and further alter the development of fat cells in fetus, resulting in increased risk for obesity throughout the lifecourse [45,46]. Furthermore, early pregnancy is the period of hypothalamus development in the offspring, which is important for appetite regulation later in life [47,48].

To our knowledge, no previous studies have assessed sex-specific associations of GWG with child weight status after birth. In our study, greater total GWG rate was positively associated with infant BMI z-score at 3 and 6 months in male but not female infants. These sex differences could be due to the differences in adiposity related hormones (such as leptin and testosterone) between male and female infants [49,50,51]. Female infants tend to have higher leptin levels than boys [49,51]. Unlike during adulthood when leptin resistance develops, in early life, greater leptin exposure likely acts to suppress appetite and may influence hypothalamic development, limiting subsequent excess weight gain. These sex differences may also be related to societal preference for sons in China, where male offspring are likely to enjoy more resources than their sisters [52]. In an over nourished environment which may lead to greater GWG, male infants are perhaps also more likely to be overfed.

The strengths of our research include a population-based prospective design, collection of weight status at multiple timepoints during pregnancy, and repeated research-standard measures of offspring length and weight status during early infancy. The study was based on the understudied Asian population, which has both high prevalence of excessive GWG and rapidly increasing pace of childhood obesity. Our research also has several limitations. First, pre-pregnancy weight status was based on self-reports and thus subject to recall bias, although previous studies have reported high correlations between self-reported and measured pre-pregnancy weight [12,22]. Second, statistical power was limited in our analyses stratified by pre-pregnancy weight status, thus the results for the underweight and overweight/obese categories should be interpreted with caution. Third, our study did not have other measures of infant body composition such as fat mass, although we plan to measure fat mass directly in our cohort when the children are older. Fourth, there may be residual confounding which we have not accounted for in this analysis. Fifth, the present study focused on the weight status of offspring during early infancy, and the long-term association is not clear now but which we will investigate in the later follow up. Lastly, our findings are based on a regional population, thus may not generalize to other settings.

## 5. Conclusions

In summary, we found that GWG rate was positively associated with early infancy weight and BMI z-score, but not with length. Normal weight women are more likely to have infants with higher BMI z-score if they had greater weight gain during pregnancy, thus they should pay more attention to control weight gain. Associations were stronger in male infants. Given that boys in China are approximately twice as likely to have obesity, and the rate of obesity is rising much faster than in girls [53], more attention should be paid to early life obesity prevention among boys in China. Furthermore, the observed associations of GWG rate with BMI z-score were stronger in the first two trimesters, which suggests that interventions to prevent excessive GWG should be conducted in early pregnancy, or perhaps even begun prior to pregnancy, to reduce the risk of childhood overweight or obesity later in life.

## Figures and Tables

**Figure 1 nutrients-11-00280-f001:**
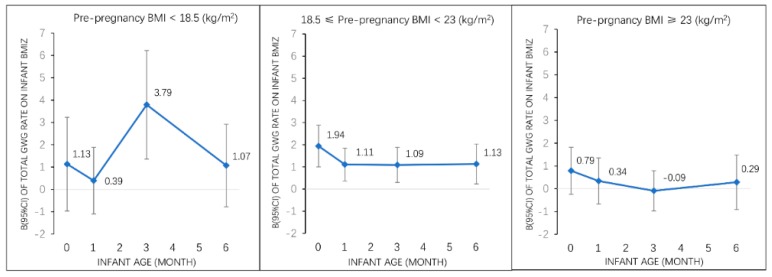
Associations of total GWG rate with infant BMI z-score (β, 95% CI) from birth to 6 months among all participants and by pre-pregnancy weight status. Adjusted for maternal pre-pregnancy BMI, maternal age, race, parity, education, household income, smoking status, paternal BMI. Abbreviations: GWG, gestational weight gain; BMI z-score, body mass index-for-age z-score; BMI, body mass index; CI, confidence interval.

**Figure 2 nutrients-11-00280-f002:**
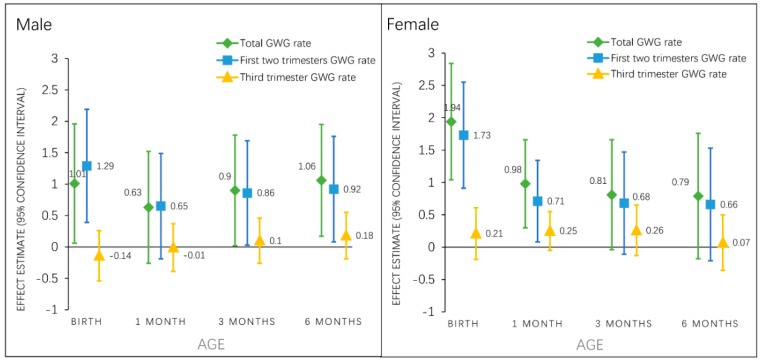
Associations of total and trimester specific GWG rate with sex-specific BMI z-score (β, 95% CI) from birth to 6 months. Total GWG rate and first two trimesters GWG rate models are adjusted for maternal pre-pregnancy BMI category, age, race, parity, education, family income, smoking status, paternal BMI category; third trimester GWG rate models are further adjusted for gestational diabetes status and the GWG rate in the first two trimesters. Abbreviations: GWG, gestational weight gain; BMIZ, body mass index-for-age z-score.

**Table 1 nutrients-11-00280-t001:** Participants characteristics according to quartiles of total gestational weight gain rate among 801 mother–infant pairs in Born in Shenyang Cohort.

Characteristic	Subjects, Mean ± SD or *n* (%)	Quartile of Gestational Weight Gain Rate, Mean ± SD or *n* (%)				
		Q1	Q2	Q3	Q4	*p*-Value
	*n* = 801	*n* = 194	*n* = 205	*n* = 202	*n* = 200	
Maternal						
Age, year	30.32 ± 3.88	30.89 ± 3.92	30.28 ± 3.93	30.08 ± 3.92	30.05 ± 3.67	0.116
Race						0.679
Han	667 (83.3)	163 (84.0)	175 (85.4)	167 (82.7)	162 (81.0)	
Others	134 (16.7)	31 (16.0)	30 (14.6)	35 (17.3)	38 (19.0)	
Educational attainment						0.256
Middle school or below	64 (8.0)	12 (6.2)	12 (5.9)	17 (8.4)	23 (11.5)	
High school	128 (16.0)	30 (15.5)	32 (15.6)	26 (12.9)	40 (20.0)	
College	533 (66.5)	134 (69.1)	141 (68.8)	136 (67.3)	122 (61.0)	
Graduate or above	76 (9.5)	18 (9.3)	20 (9.8)	23 (11.4)	15 (7.5)	
Household income per year, CNY						0.773
<30,000	200 (25.0)	46 (23.7)	51 (24.9)	55 (27.2)	48 (24.0)	
30,000–<50,000	229 (28.6)	49 (25.3)	66 (32.2)	53 (26.2)	61 (30.5)	
50,000–<70,000	170 (21.2)	46 (23.7)	45 (22.0)	40 (19.8)	39 (19.5)	
≥70,000	202 (25.2)	53 (27.3)	43 (21.0)	54 (26.7)	52 (26.0)	
Nulliparous	610 (76.2)	148 (76.3)	159 (77.6)	155 (76.7)	148 (74)	0.856
Gestational diabetes	159 (20.0)	56 (28.9)	41 (20.0)	31 (15.4)	31 (15.5)	0.002
Smoking status						0.171
Never	764 (95.4)	182 (93.8)	198 (96.6)	196 (97.0)	188 (94.0)	
Former	34 (4.2)	12 (6.2)	7 (3.4)	4 (2.0)	11 (5.5)	
During pregnancy	3 (0.3)	0 (0.0)	0 (0.0)	2 (1.0)	1 (0.5)	
Pre-pregnancy BMI, kg/m^2^	22.04 ± 3.52	23.41 ± 4.12	21.78 ± 3.37	21.39 ± 3.24	21.64 ± 2.97	<0.001
Pre-pregnancy BMI category						<0.001
<18.5, kg/m^2^	113 (14.1)	21 (10.8)	29 (14.2)	36 (17.8)	27 (13.5)	
18.5–<24.0, kg/m^2^	442 (52.7)	80 (41.2)	116 (56.6)	113 (56.0)	113 (56.5)	
≥24.0, kg/m^2^	266 (33.2)	93 (47.9)	60 (29.3)	53 (26.2)	60 (30.0)	
Gestation length, wk	38.85 ± 1.16	38.74 ± 1.19	38.90 ± 1.08	38.85 ± 1.24	38.89 ± 1.14	0.524
1st and 2nd trimesters GWG rate, kg/wk	0.34 ± 0.14	0.19 ± 0.09	0.30 ± 0.07	0.38 ± 0.08	0.50 ± 0.12	<0.001
3rd trimester GWG rate, kg/wk	0.71 ± 0.30	0.50 ± 0.21	0.65 ± 0.19	0.74 ± 0.23	0.93 ± 0.37	<0.001
Paternal						
BMI, kg/m^2^	25.08 ± 3.75	25.29 ± 3.71	24.86 ± 3.82	24.85 ± 3.54	25.32 ± 3.91	0.405
Infant						
Female	419 (52.3)	116 (59.8)	101 (49.3)	105 (52.0)	97 (48.5)	0.098
Birth weight, gm	3401 ± 453	3292 ± 442	3342 ± 424	3423 ± 441	3548 ± 469	<0.001

Abbreviations: SD, Standard deviation; CNY, China Yuan (1 China Yuan = 0.14 US Dollar); BMI, body mass index; GWG, gestational weight gain.

**Table 2 nutrients-11-00280-t002:** Associations of total and trimester-specific gestational weight gain rate with infant growth measures from birth to 6 months using linear mixed effects model.

Infant Growth Measures	Total GWG Rate (kg/wk)	1st and 2nd Trimesters GWG Rate (kg/wk)	3rd Trimester GWG Rate (kg/wk)
	β (95% CI), (*n* = 801)		
BMIZ			
Model 1	1.16 (0.67, 1.66)	1.11 (0.65, 1.57)	0.10 (−0.12, 0.32)
Model 2	1.36 (0.86, 1.85)	1.28 (0.82, 1.75)	0.12 (−0.10, 0.34)
Model 3	1.34 (0.84, 1.83)	1.31 (0.85, 1.77)	0.08 (−0.13, 0.30) ^a^
WFAZ			
Model 1	0.98 (0.54, 1.41)	0.88 (0.47, 1.29)	0.07 (−0.13, 0.26)
Model 2	1.17 (0.73, 1.61)	1.05 (0.64, 1.46)	0.09 (−0.10, 0.29)
Model 3	1.18 (0.75, 1,62)	1.08 (0.76, 1.49)	0.14 (−0.08, 0.36) ^a^
LFAZ			
Model 1	0.33 (−0.22, 0.88)	0.26 (−0.26, 0.78)	−0.05 (−0.29, 0.20)
Model 2	0.44 (−0.12, 0.99)	0.36 (−0.16, 0.88)	−0.03 (−0.28, 0.21)
Model 3	0.48 (−0.07, 1.04)	0.39 (−0.13, 0.91)	0.06 (−0.13, 0.26) ^a^

Model 1: adjusted for exact age of infants at each measurement; Model 2: Model 1 + pre-pregnancy body mass index; Model 3: Model 2 + maternal age, race, parity, education, household income, smoking status, paternal body mass index; ^a^: Further adjusted 1st and 2nd trimesters GWG rate, gestational diabetes; BMIZ, body mass index-for-age and sex z-score; WFAZ, weight-for-age z-score; LFAZ, length-for-age z-score.

**Table 3 nutrients-11-00280-t003:** Associations of total and trimester-specific gestational weight gain rate with infant growth measures at birth, 1 month, 3 months, and 6 months among 801 mother–infant pairs in Born in Shenyang Cohort.

Infant Growth Measures	Total GWG Rate (kg/wk)	1st and 2nd Trimesters GWG Rate (kg/wk)	3rd Trimester GWG Rate ^a^ (kg/wk)
	β (95% CI), (*n* = 801)		
BMIZ			
0 month	1.56 (0.91, 2.21)	1.62 (1.02, 2.22)	0.03 (−0.25, 0.30)
1 month	0.86 (0.31, 1.41)	0.71 (0.20, 1.22)	0.14 (−0.10, 0.38)
3 months	0.99 (0.40, 1.59)	0.78 (0.21, 1.34)	0.20 (−0.06, 0.47)
6 months	0.82 (0.16, 1.47)	0.70 (0.09, 1.32)	0.11 (−0.17, 0.39)
WFAZ			
0 month	1.42 (0.94, 1.91)	1.36 (0.91, 1.81)	0.05 (−0.17, 0.26)
1 month	0.79 (0.34, 1.24)	0.68 (0.26, 1.11)	0.05 (−0.15, 0.25)
3 months	0.83 (0.32, 1.34)	0.74 (0.27, 1.22)	0.05 (−0.17, 0.28)
6 months	0.75 (0.18, 1.31)	0.87 (0.34, 1.39)	−0.07 (−0.32, −0.17)
LFAZ			
0 month	0.78 (0.09, 1.47)	0.60 (−0.05, 1.24)	0.04 (−0.26, 0.33)
1 month	0.38 (−0.23, 1.00)	0.39 (−0.18, 0.96)	−0.11 (−0.38, 0.16)
3 months	0.19 (−0.42, 0.80)	0.31 (−0.27, 0.88)	−0.16 (−0.43, 0.11)
6 months	0.22 (−0.42, 0.86)	0.59 (−0.01, 1.18)	−0.28 (−0.57, 0.01)

Adjusted for pre-pregnancy body mass index, maternal age, race, parity, education, household income, smoking status, paternal body mass index. ^a^: Further adjusted 1st and 2nd trimesters GWG rate, gestational diabetes; BMIZ, body mass index-for-age and sex z-score; WFAZ, weight-for-age z-score; LFAZ, length-for-age z-score.

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
