# Peer review of "Association of Total and Trimester-Specific Gestational Weight Gain Rate with Early Infancy Weight Status: A Prospective Birth Cohort Study in China"

_nutrients, 2019, doi:10.3390/nu11020280_

Round 1

Reviewer 1 Report

This is an interesting a pre-birth cohort study with follow up to 6 months of age, aimed to assess if: greater GWG rate was positively associated with offspring weight; first and second trimesters weight gain had stronger associations with offspring weight; and the association of GWG with offspring weight differed by infant sex or pre-pregnancy BMI. The assumption that GWG patterns in Chinese population might differ from that reported in Western populations was a rationale for the study. Infant BMI z-score was used as a surrogate of adiposity.

Most results are consistent with those from literature, such as the positive association between GWG and offspring weight and adiposity (Hull 2011, Goldstein 2017), early rather than late GWG positively associated with offspring weight, and a greater effect of maternal adiposity in boys (Pereira-da-Silva 2014, Andres 2015).

Queries:

The aforementioned results were presented but the possible mechanisms for some associations were not discussed. For instance, why the greater GWG rate associated with offspring BMI z-score is mostly driven by early instead of late GWG? Why this association affects predominantly male offspring?

Surprisingly, greater GWG rate was found to be significantly associated with lower prevalence of both GDM and pre-pregnancy overweight/obesity. These results are in contrast with other studies conducted in Chinese populations reporting positive associations of GWG both with GDM (Chen 2010) and pre-pregnancy overweight/obesity (Zhao 2018). In addition, no association was found between total GWG rate and offspring BMI z-score among overweight/obese mothers, also is in contrast with other study reporting that the association between offspring obesity and GWG is stronger among mothers who were overweight prior to pregnancy than those who had normal weight (Suebe 2009). Findings of this study that are contradictory with those from other studies should be adequately compared and discussed.

References

- Andres A, Hull HR, Shankar K, Casey PH, Cleves MA, Badger TM. Longitudinal body composition of children born to mothers with normal weight, overweight, and obesity. Obesity (Silver Spring). 2015;23:1252-8.

Chen Z, Du J, Shao L, Zheng L, Wu M, Ai M, et al. Prepregnancy body mass index, gestational weight gain, and pregnancy outcomes in China. Int J Gynaecol Obstet. 2010;109:41-4

- Goldstein RF, Abell SK, Ranasinha S, Misso M, Boyle JA, Black MH, et al. Association of gestational weight gain with maternal and infant outcomes: a systematic review and meta-analysis. JAMA. 2017;317:2207-25.

- Hull HR, Thornton JC, Ji Y, Paley C, Rosenn B, Mathews P, et al. Higher infant body fat with excessive gestational weight gain in overweight women. Am J Obstet Gynecol 2011;205:211.e1-7.

- Pereira-da-Silva L, Cabo C, Moreira AC, Virella D, Guerra T, Camoes et al. The adjusted effect of maternal body mass index, energy and macronutrient intakes during pregnancy, and gestational weight gain on body composition of full-term neonates. Am J Perinatol. 2014;31:875-82.

Stuebe AM, Forman MR, Michels KB. Maternal-recalled gestational weight gain, pre-pregnancy body mass index, and obesity in the daughter. Int J Obes (Lond). 2009;33:743-52.

Author Response

Response to Reviewer 1 Comments

Point 1: This is an interesting a pre-birth cohort study with follow up to 6 months of age, aimed to assess if: greater GWG rate was positively associated with offspring weight; first and second trimesters weight gain had stronger associations with offspring weight; and the association of GWG with offspring weight differed by infant sex or pre-pregnancy BMI. The assumption that GWG patterns in Chinese population might differ from that reported in Western populations was a rationale for the study. Infant BMI z-score was used as a surrogate of adiposity. Most results are consistent with those from literature, such as the positive association between GWG and offspring weight and adiposity (Hull 2011, Goldstein 2017), early rather than late GWG positively associated with offspring weight, and a greater effect of maternal adiposity in boys (Pereira-da-Silva 2014, Andres 2015).

Response 1: We thank Reviewer 1 for the positive evaluation of our study.

Point 2: The aforementioned results were presented but the possible mechanisms for some associations were not discussed. For instance, why the greater GWG rate associated with offspring BMI z-score is mostly driven by early instead of late GWG? Why this association affects predominantly male offspring?

Response 2: Greater GWG in early pregnancy may reduce maternal insulin sensitivity and glucose tolerance which could expose the fetus to an increased glucose supply and further alter the development of fat cells in fetus, resulting in increased risk for obesity throughout the lifecourse. Furthermore, early pregnancy is the period of hypothalamus development in the offspring, which is important for appetite regulation later in life.

The observed sex differences could be due to the differences in adiposity related hormones (such as leptin and testosterone) between male and female infants. Female infants tend to have higher leptin levels than boys. Unlike during adulthood when leptin resistance develops, in early life, greater leptin exposure likely acts to suppress appetite and may influence hypothalamic development, limiting subsequent excess weight gain. These sex differences may also be related to societal preference for sons in China, where male offspring are likely to enjoy more resources than their sisters. In an over nourished environment which may lead to greater GWG, male infants are perhaps also more likely to be overfed.

We have included these statements in the Discussion on page 9-10, paragraph 4-5, lines 278-293

Point 3: Surprisingly, greater GWG rate was found to be significantly associated with lower prevalence of both GDM and pre-pregnancy overweight/obesity. These results are in contrast with other studies conducted in Chinese populations reporting positive associations of GWG both with GDM (Chen 2010) and pre-pregnancy overweight/obesity (Zhao 2018). In addition, no association was found between total GWG rate and offspring BMI z-score among overweight/obese mothers, also is in contrast with other study reporting that the association between offspring obesity and GWG is stronger among mothers who were overweight prior to pregnancy than those who had normal weight (Suebe 2009). Findings of this study that are contradictory with those from other studies should be adequately compared and discussed.

Response 3: We thank Reviewer 1 for the valuable comments. Findings of our study are contradictory with some of previous study, however consistent with the other studies, which we should discuss. Please see page 9, paragraph 2, lines 250-258

Response to Reviewer 2 Comments

Point 1: The current study investigates longitudinal relations between mothers’ gestational weight gain and infant weight at different time points within the first year of life. The study is well written and the hypotheses are logical and well structured. The sample size is adequate and the methodology (longitudinal study) very suitable for the research question. In general, the study seems useful for society and very suitable for the journal.

Response 1: We thank Reviewer 2 for the positive evaluation of our study.

Point 2: Analyses: The authors test numerous different models. Were the findings corrected for multiple comparisons?

Response 2: Thank you for the comment. Given all outcomes are related in the present study, the authors think it is reasonable for not conducting multiple testing (Aris 2019), so we did not test for multiple comparisons.

Point 3: l. 168 ff.: The authors report that those participants with higher GWG had a lower prevalence of pregnancy obesity. The results suggest that participants who were overweight during pregnancy also gained less weight. Therefore, the smaller effects of GWG in obese participants may be due to the fact that they gained less weight in general, which could be discussed.

Response 3: Thank you for the valuable comments. That helps to explain why the association between GWG and offspring weight status was not significant among pregnancies who were overweight/obese before pregnancy, we will add this into the discussion. Please see page 9, paragraph 2, lines 254-258

Point 4: Furthermore, the authors should test whether there is a general relationship of maternal weight and infant weight/ BMI in the current study. Obese mothers may generally have heavier children, leading to a ceiling effect and therefore no relation between GWG and child BMI in obese mothers. 

Response 4: We further tested the relationship between pre-pregnancy BMI (Supplementary material: Table S3) and add this comment into the discussion. Please see page 9, paragraph 2, lines 254-258

Point 5: The relation between GWG and infant weight seems to decrease with infant age. It should be discussed whether this indicates that GWG of the mother only has a short-term effect on infant weight that will not last beyond the first year of life. A further follow up session might have shed further light on this issue. However, the study is already interesting in its current state.

Response 5: Thank you for the comments. We will investigate the association in the later follow up and have addressed this point in the Discussion (page 10, lines 305-307).

Point 6: A large number of abbreviations is used. To increase readability, I suggest not abbreviating those words that are only infrequently used (e.g. BMIZ, WFSZ, LFAZ, GDM).

Response 6: Thank you for the suggestion. We will try not to use those abbreviations to increase the readability except in abstract which limited by the word number of 200.

Point 7: L. 233 “our study is the first study to assessed associations of trimester-specific GWG” – this is a typo and should read “our study is the first study to assess associations of trimester-specific GWG”

Response 7: Thank you, it’s a mistake and we’ve revised it.

Reviewer 2 Report

The current study investigates longitudinal relations between mothers’ gestational weight gain and infant weight at different time points within the first year of life. The study is well written and the hypotheses are logical and well structured. The sample size is adequate and the methodology (longitudinal study) very suitable for the research question. In general, the study seems useful for society and very suitable for the journal. I only have some minor suggestions:

Analyses: The authors test numerous different models. Were the findings corrected for multiple comparisons?

l. 168 ff.: The authors report that those participants with higher GWG had a lower prevalence of pregnancy obesity. The results suggests that participants who were overweight during pregnancy also gained less weight. Therefore, the smaller effects of GWG in obese participants may be due to the fact that they gained less weight in general, which could be discussed.

Furthermore, the authors should test whether there is a general relationship of maternal weight and infant weight/ BMI in the current study. Obese mothers may generally have heavier children, leading to a ceiling effect and therefore no relation between GWG and child BMI in obese mothers.  

The relation between GWG and infant weight seems to decrease with infant age. It should be discussed whether this indicates that GWG of the mother only has a short-term effect on infant weight that will not last beyond the first year of life. A further follow up session might have shed further light on this issue. However, the study is already interesting in its current state.

A large number of abbreviations is used. To increase readability, I suggest not abbreviating those words that are only infrequently used (e.g. BMIZ, WFSZ, LFAZ, GDM).

L. 233 “our study is the first study to assessed associations of trimester-specific GWG” – this is a typo and should read “our study is the first study to assess associations of trimester-specific GWG

Author Response

Response to Reviewer 2 Comments

Point 1: The current study investigates longitudinal relations between mothers’ gestational weight gain and infant weight at different time points within the first year of life. The study is well written and the hypotheses are logical and well structured. The sample size is adequate and the methodology (longitudinal study) very suitable for the research question. In general, the study seems useful for society and very suitable for the journal.

Response 1: We thank Reviewer 2 for the positive evaluation of our study.

Point 2: Analyses: The authors test numerous different models. Were the findings corrected for multiple comparisons?

Response 2: Thank you for the comment. Given all outcomes are related in the present study, the authors think it is reasonable for not conducting multiple testing (Aris 2019), so we did not test for multiple comparisons.

Point 3: l. 168 ff.: The authors report that those participants with higher GWG had a lower prevalence of pregnancy obesity. The results suggest that participants who were overweight during pregnancy also gained less weight. Therefore, the smaller effects of GWG in obese participants may be due to the fact that they gained less weight in general, which could be discussed.

Response 3: Thank you for the valuable comments. That helps to explain why the association between GWG and offspring weight status was not significant among pregnancies who were overweight/obese before pregnancy, we will add this into the discussion. Please see page 9, paragraph 2, lines 254-258

Point 4: Furthermore, the authors should test whether there is a general relationship of maternal weight and infant weight/ BMI in the current study. Obese mothers may generally have heavier children, leading to a ceiling effect and therefore no relation between GWG and child BMI in obese mothers. 

Response 4: We further tested the relationship between pre-pregnancy BMI (Supplementary material: Table S3) and add this comment into the discussion. Please see page 9, paragraph 2, lines 254-258

Point 5: The relation between GWG and infant weight seems to decrease with infant age. It should be discussed whether this indicates that GWG of the mother only has a short-term effect on infant weight that will not last beyond the first year of life. A further follow up session might have shed further light on this issue. However, the study is already interesting in its current state.

Response 5: Thank you for the comments. We will investigate the association in the later follow up and have addressed this point in the Discussion (page 10, lines 305-307).

Point 6: A large number of abbreviations is used. To increase readability, I suggest not abbreviating those words that are only infrequently used (e.g. BMIZ, WFSZ, LFAZ, GDM).

Response 6: Thank you for the suggestion. We will try not to use those abbreviations to increase the readability except in abstract which limited by the word number of 200.

Point 7: L. 233 “our study is the first study to assessed associations of trimester-specific GWG” – this is a typo and should read “our study is the first study to assess associations of trimester-specific GWG”

Response 7: Thank you, it’s a mistake and we’ve revised it.
